# N-heterocyclic carbene-catalyzed atroposelective synthesis of *N*-Aryl phthalimides and maleimides via activation of carboxylic acids

Soumen Barik[1], Sowmya Shree Ranganathappa[1] & Akkattu T. Biju [1] ✉

Traditionally, *N*-aryl phthalimides are synthesized by the condensation of phthalic anhydride and aniline derivatives, usually proceeding under harsh conditions. The alternative mild and organocatalytic strategies for their synthesis are underdeveloped. Herein, we demonstrate the organocatalytic atroposelective synthesis of *N*-aryl phthalimides via the traditional N-$C_{C=O}$ disconnection under mild conditions. The in-situ acid activation of phthalamic acid and subsequent N-heterocyclic carbene (NHC)-catalyzed atroposelective amidation allowed the synthesis of well-decorated *N*-aryl phthalimides in excellent yields and enantioselectivities. Mechanistic studies reveal the addition of NHC to the in situ generated isoimides, thus introducing a unique mode of generating acylazoliums. Interestingly, both enantiomers of the product can be accessed from the same phthalic anhydride and aniline using the same NHC pre-catalyst. Moreover, this strategy has been extended to the atroposelective synthesis of *N*-aryl maleimides.

*N*-Substituted phthalimides are ubiquitous in numerous bioactive compounds[1,2]. For instance, thalidomide having phthalimide skeleton is used for the treatment of multiple myeloma[3] and tuberculosis[4], while apremilast is useful for the treatment of psoriatic arthritis[5]. Moreover, *N*-substituted phthalimide derivatives are widely applicable as catalysts[6,7], dyes[8], and for the synthesis of polymers[9]. Given the diverse applications of phthalimides, direct and mild synthesis of such molecules is highly desirable.

Among the various strategies known for the construction of *N*-substituted phthalimides[10,11], the most common approach involves the condensation of phthalic anhydride and primary amine at elevated temperature or in the presence of acidic dehydrating agents (Fig. 1A)[12–14]. Moreover, a variety of transition metal-catalyzed synthesis of functionalized *N*-substituted phthalimides are known, such as the carbonylative cyclization of aromatic amides[15–17] or 1,2-dihaloarenes[18] in the presence of CO at high pressure. *N*-substituted phthalimides can also be synthesized from diols via Ru catalysis[19]. or the cyclization of isocyanates with benzoic acid/amide derivatives[20,21]. However, all these

methods require pre-functionalized starting materials, high temperatures, and/or expensive metal catalysts to furnish the phthalimides. Additionally, the atroposelective synthesis of *N*-aryl phthalimides via Pd-catalyzed carbonylation of aryl iodide with CO was disclosed by Li group[22]. Notably, very few organocatalytic routes for the synthesis of *N*-aryl phthalimides have been realized. The benzannulation strategy via oxidative [4 + 2] annulation of α,β-unsaturated aldehydes with *N*-substituted maleimides for the synthesis of achiral[23] and atroposelective[24] *N*-substituted phthalimides has been established. However, these methodologies require pre-functionalized maleimides as starting materials for the [4 + 2] annulation.

Although the dehydration of phthalamic acid needs drastic reaction conditions (Fig. 1A), this is a straightforward route to phthalimides from the two readily available starting materials. However, the use of this disconnection for the atroposelective synthesis of *N*-aryl phthalimides has not been hitherto accomplished. We hypothesized that phthalamic acid generated from phthalic anhydride and 2-substituted aniline could be activated by using pivaloyl chloride (PivCl), and this

[1]Department of Organic Chemistry, Indian Institute of Science, Bangalore 560012, India. ✉e-mail: atbiju@iisc.ac.in

**Fig. 1 | Traditional N−C$_{C=O}$ disconnection for *N*-aryl phthalimides synthesis.**
**A** Traditional method for the synthesis of achiral *N*-substituted phthalimides. **B** This work: atroposelective synthesis of *N*-aryl phthalimides/maleimides.

could possibly generate the isoimide intermediate[25]. Thereafter, the addition of a chiral N-heterocyclic carbene (NHC)[26–41] to the isoimide could generate the acylazolium intermediate, which make the carbonyl carbon electrophilic enough for the atroposelective amidation of less nucleophilic *N*-aryl acid amides (Fig. 1B)[42–45]. Based on this concept, herein, we report the organocatalytic atroposelective synthesis of *N*-aryl phthalimides via the traditional N−C$_{C=O}$ disconnection under mild conditions[46–48]. Notably, the NHC-catalyzed activation of carboxylic acids for the enantioselective synthesis of heterocycles was independently reported by the Scheidt[49], and Ye groups[50]. Although this strategy has been subsequently used by many groups for the enantioselective synthesis of a plethora of chiral compounds[51–54], atroposelective synthesis using the acid activation employing NHC catalysis has not been explored to date[55–61]. It is worth mentioning that, related atroposelective amidation strategy has been recently employed by our group for the synthesis of N−N axially chiral quinazolinones[62].

## Results

### Reaction optimization for C-N axially chiral phthalimides
Our present study commenced with the treatment of substituted benzoic acid **1a** with PivCl and the carbene generated from the chiral pre-catalyst **3**[63] using K$_2$CO$_3$ in THF. Under these conditions, we were elated to obtain the enantioenriched C-N axially chiral phthalimide **2a** in 99% yield and 98:2 enantiomeric ratio (er; Table 1, entry 1). The screening of NHC catalysts revealed that carbenes derived from the triazolium salts **4** and **6** were equally effective for catalyzing the amidation reaction; however, the carbene precursor **5** was found to be ineffective (entries 2–4). The reaction did not work without the acid activation using PivCl (entry 5), and the use of HATU as the acid activator instead of PivCl resulted in reduced yield and er of **2a** (entry 6)[53]. The use of other (in) organic bases such as Cs$_2$CO$_3$, Na$_2$CO$_3$, DMAP and DBU furnished reduced er of **2a** although the reactivity was good (entries 7-10). Moreover, the reactions performed in other solvents such as MTBE, CHCl$_3$ and toluene returned inferior results compared to the standard conditions (entries 11-13). The reaction run for 12 h instead of 24 h

afforded **2a** in reduced yield of 82% maintaining the high selectivity (entry 14). Hence, entry 1 in Table 1 was selected for the substrate scope analysis (For details, please see the Supplementary Sections 1.3, 1.4).

### Substrate scope and studies on mechanism
With the identified reaction conditions in hand, the generality of the reaction was then investigated. Initially, the phthalamic acids derived from electronically different anilines were tested (Fig. 2). A series of 2-*tert* butyl aniline-derived phthalamic acids bearing electron-donating substituents, halogens, and aryl group at the *para*-position underwent smooth atroposelective amidation reaction under the optimized conditions to furnish the corresponding *N*-aryl phthalimides in excellent yields and high er values (**2b**-**2g**). The electron withdrawing ester moiety was also well tolerated affording **2 h** in 97% yield and 92:8 er. The presence of thienyl moiety, and carbon-carbon multiple bonds at the 4-position of aniline did not adversely affect the outcome of the reaction (**2i**-**2l**). Moreover, phthalamic acid derived from the *meta*-nitro substituted aniline afforded the desired product **2 m** in 99% yield and 97:3 er. We thereafter searched the other substitutions in the *ortho* position of anilines. Replacing the *tert*-butyl group at the *ortho* position of aniline to a cumene group or a methoxy methyl propyl group yielded the corresponding products **2n** and **2o** without a noticeable change in the er. The incorporation of di-aryl or di-heteroaryl moieties effectively restrains rotation around the C-N bond, thereby facilitating the formation of **2p** and **2q** with excellent enantioselectivity. The *N*-aryl moiety, featuring a phenyl sulfonyl group at the *ortho* position, poses a challenge in rotational control around the C-N axis, owing to the elongated C−S bond compared to that in the C−C (bulky alkyl group) bond. Despite this challenge, the synthesis of **2r** was accomplished with 71% yield and 80:20 er. Notably, the reactions performed using di-*ortho* substituted aniline-derived phthalamic acid substrates although reacted well but with poor er values. Next, we evaluated the variation in the benzoic acid moiety. Electron-donating groups and halo substitution at the *ortho*-position of phthalamic acids provided the target *N*-aryl phthalimides with excellent yields and enantioselectivities (**2s**-**2x**).

The structure and the absolute stereochemistry of the C−N axis was confirmed by the X-ray analysis of compounds **2w** and **2x** (CCDC 2252546 (**2w**) and CCDC 2252545 (**2x**)). The absolute configuration of the C−N axis of other phthalimides was assigned by analogy. Notably, the 2-nitro substituted phthalamic acid provided the product **2 y** in 99% yield and 88:12 er. Interestingly, scope of the present methodology could be expanded beyond the atroposelective synthesis of monosubstituted phthalimides. Phthalamic acids generated from di- and tri-substituted phthalic anhydrides also proved to be effective substrates, resulting in the synthesis of di- and tri-substituted phthalimides in excellent yields and enantioselectivities. Unsymmetrical naphthyl phthalic anhydride-derived phthalimide **2z**, was formed in 95% yield with 96:4 er. Similarly, disubstituted phthalimides such as 4-methoxy-5-nitro phthalimide **2aa** and 4-methoxy-7-nitro phthalimide **2ab** were formed in excellent yields with good to excellent enantioselectivities. Moreover, sterically demanding tri-substituted phthalimic acid did not affect the reaction outcome (**2ac**). Furthermore, performing the reaction with the acid derived from pyridinic anhydride afforded the desired product **2ad** in 99% yield and 71:29 er, proving the generality of the present methodology.

The reaction of unsymmetrical phthalic anhydride with aniline derivatives generally produces two regioisomeric phthalamic acids, where one regioisomer forms predominantly due to the selective amine addition to the less sterically hindered carbonyl. For instance, the treatment of 4-methylisobenzo furan-1,3-dione with 2-*tert*-butyl aniline afforded the phthalamic acids **1a** and **1a′** as separable mixture of regioisomers in 56% and 12% isolated yields respectively (Fig. 3A).

**Table 1 | Optimization of the reaction conditions[a]**

| entry | variation of the standard conditions[a] | yield of 2a (%)[b] | er of 2a[c] |
|---|---|---|---|
| 1 | none | 99 (99)[d] | 98:2 |
| 2 | **4** instead of **3** | 99 | 97:3 |
| 3 | **5** instead of **3** | <5 | ND |
| 4 | **6** instead of **3** | 96 | 5:95 |
| 5 | without PivCl | <5 | ND |
| 6 | HATU instead of PivCl | 92 | 87:13 |
| 7 | Cs$_2$CO$_3$ instead of K$_2$CO$_3$ | 99 | 89:11 |
| 8 | Na$_2$CO$_3$ instead of K$_2$CO$_3$ | 90 | 92:8 |
| 9 | DMAP instead of K$_2$CO$_3$ | 75 | 81:19 |
| 10 | DBU instead of K$_2$CO$_3$ | 90 | 87:13 |
| 11 | MTBE instead of THF | 91 | 93:7 |
| 12 | CHCl$_3$ instead of THF | 80 | 77:23 |
| 13 | toluene instead of THF | 70 | 96:4 |
| 14 | 12 h instead of 24 h | 82 | 98:2 |

[a]Standard conditions: **1a** (0.125 mmol), **3** (10 mol %), K$_2$CO$_3$ (1.5 equiv), THF (1.5 mL), 25 °C and 24 h.
[b]Determined by $^1$H NMR analysis of crude products using CH$_2$Br$_2$ as the internal standard.
[c]The er value was determined by HPLC analysis on a chiral stationary phase.
[d]Isolated yield.

The structure of regioisomer **1a** was confirmed using X-ray analysis of the crystals (CCDC 2261808 (**1a**)).

Interestingly, both enantiomers of the C-N axially chiral *N*-aryl phthalimide can be accessed using the same phthalic anhydride and aniline employing the same enantiomer of the NHC pre-catalyst. Treatment of **1a** under the optimized conditions afforded **2a** in 99% yield and 98:2 er. The reaction of the regioisomer **1a'** with the same NHC pre-catalyst **3** furnished the enantiomer of **2a** (*ent*-**2a**) in 99% yield and 3:97 er. Thus, by tuning the substitution position on benzoic acid moiety, the opposite enantiomer is accessible.

To gain insight into the rotational restriction around the C–N bond, the *N*-aryl phthalimide **2a** was heated for 2 h at different temperatures in toluene and monitored the change in ee values. Notably, up to 70 °C, there was no change in ee value revealing the restricted rotation around the C–N bond (Fig. 3B). Further, an increase in temperature beyond 70 °C allowed the rotation around the C–N axis as observed by the lowering of ee values, and at 140 °C the ee was 0% indicating complete racemization. For compound **2n**, the rotational restriction around the C–N bond was observed up to 50 °C, and the complete racemization was observed at 120 °C. We have also determined the C–N rotational barrier for compounds **2a** and **2n** experimentally and with the aid of density functional theory (DFT) studies. By monitoring the variation of er values at different time intervals while keeping the temperature at 100 °C, the $\Delta G_{rot}^{\ddagger}$ for the C–N bond in **2a** was determined experimentally to be (30.46 ± 0.03) kcal/mol using the procedure of Curran[64]. The DFT calculated value was 31.9 kcal/mol,

which is in good agreement with the experimental value. Similarly, the experimental C–N rotational barrier for **2n** was (29.35 ± 0.04) kcal/mol, and the calculated value was 30.2 kcal/mol. Moreover, the t$_{1/2}$ of racemization for **2a** and **2n** were determined to be 35.8 years and 5.5 years respectively at 25 °C.

The present catalytic strategy is scalable in a 1.0 mmol scale without erosion of reactivity and selectivity. The reaction of **1a** under the optimized conditions furnished **2a** in 99% yield and 98:2 er (Fig. 4A). Moreover, lowering the catalyst loading to 2.0 mol % of **3**, the desired products **2a** and **2t** were synthesized in comparable yields and selectivities (Fig. 4B), thereby demonstrating the practicality of the developed protocol.

To gain insight into the mechanism of the reaction, a few experiments were performed. When the phthalamic acid **1a** was treated with PivCl under the optimized reaction conditions in the absence of the NHC precursor, the isoimide **7a** was formed in 99% yield (Fig. 5A)[25]. Interestingly, when the isolated isoimide **7a** was treated with NHC precursor and base, the desired phthalimide **2a** was formed in 98% yield and 96:4 er (Fig. 5B). These experiments clearly indicate that the isoimide **7a**, is the intermediate in the present reaction (For details, please see Supplementary Sections 2.1(a), 2.1(b)). These studies revealed that the crucial acylazolium intermediate was formed through the addition of NHC to the isoimide intermediate. To the best of our knowledge, such NHC addition to isoimide has not been documented previously, thus paving a unique pathway for the generation of the acylazolium intermediate[26–41].

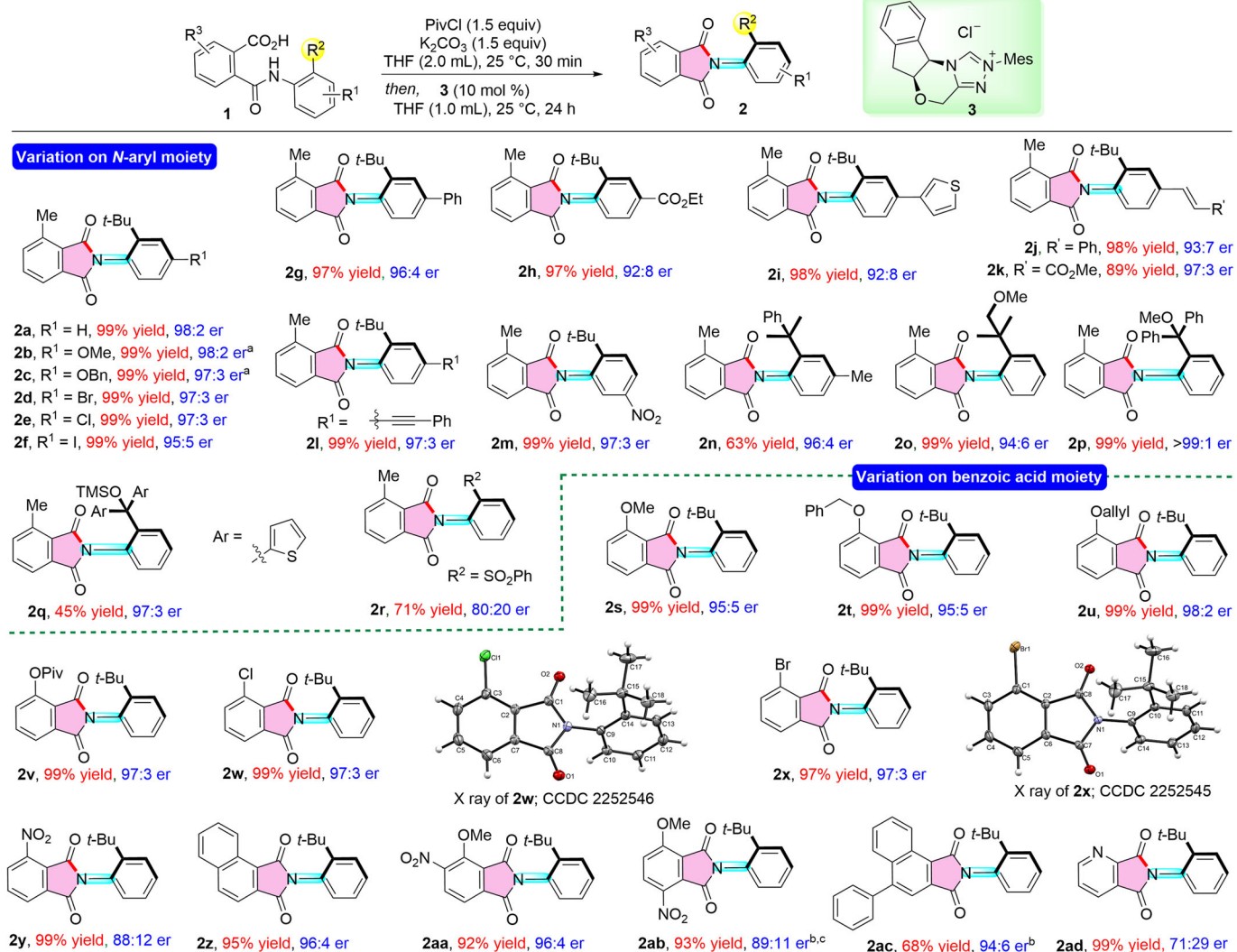

**Fig. 2 | Substrate scope for the synthesis of C-N axially chiral phthalimides.** General conditions: **1** (0.25 mmol), **3** (10 mol %), PivCl (1.5 equiv), K$_2$CO$_3$ (1.5 equiv), THF (3.0 mL), 25 °C, and 24 h. Yields of isolated products are given and the er was established by HPLC analysis on a chiral stationary phase. [a] 15 mol % **3** and 2.5 equiv of K$_2$CO$_3$ were used. [b] Reaction performed in 0.125 mmol scale. [c] DABCO was used as base and reaction performed at −20 °C.

Based on the results of the mechanistic experiments, a tentative mechanism of the reaction has been proposed (Fig. 6). The reaction likely proceeds via the in-situ activation of phthalamic acid **1a** in the presence of PivCl to form activated anhydride (**I**), which subsequently undergoes an intramolecular cyclization leading to the formation of the isoimide **7a**. The nucleophilic addition of the carbene generated from **3** to the isoimide could generate the acylazolium intermediate (**II**), which upon atroposelective amidation followed by regeneration of NHC furnished the desired C-N axially chiral phthalimides **2a**. Based on the absolute configuration determined using X-ray analysis, a plausible mode of enantioinduction has been proposed. In **Int-I**, the bulky aminoindanol and *tert*-butyl groups lie on opposite sides of the plane defined by the phthalamic acid moiety, thereby having lower energy than the sterically congested **Int-II**. Hence, the reaction is likely proceeding via the **Int-I**.

Moreover, control experiments indicated that stirring the acid **2a** in THF in the absence of **3** and K$_2$CO$_3$ afforded **2a** (racemic) in 52% yield (For details, please see the Supplementary Section 2.1(d)). The in situ generated HCl (during the pivaloyl anhydride formation, intermediate **I**, Fig. 6) could mediate the isoimide **7a** to phthalimide **2a** conversion. This was further confirmed by the direct conversion of isoimide **7a** to phthalimide **2a** in 58% yield by treatment with HCl in dioxane.

## Atroposelective synthesis of C-N axially chiral maleimides

The present NHC-catalyzed atroposelective strategy can easily be extended to the atroposelective synthesis of *N*-aryl maleimides (Fig. 7). The synthesis of axially chiral *N*-aryl amino maleimides was recently disclosed by our group[60], but this strategy involved the kinetic resolution of pre-functionalized maleimides via [3 + 3] annulation strategy in the presence of chiral NHC precursor, where a maximum 50% yield of the maleimides was accomplished. Herein, with the slightly modified reaction conditions, the *N*-aryl maleimide **9a** was formed in 65% yield with 93:7 er. Halo substitution (**9b, 9c**), as well as the electron-withdrawing CO$_2$Et group (**9d**) present in the *para*-position of aniline afforded the atroposelective amidation product in good yields and er values. Moreover, cumene moiety present at the *ortho*-position of aniline also furnished the desired maleimide **9e** without a notable change in er. Moreover, the present methodology also well effective for the atroposelective synthesis of disubstituted maleic anhydrides **9f** in good yield with moderate er value.

## Synthetic applications

The C-N axially chiral phthalimide **2a** and maleimide **9a** can be utilized as synthetically useful precursors for further synthetic elaborations of C-N axially chiral derivatives. Benzylic bromination of **2a** in presence of NBS/benzoyl peroxide leads to the synthesis of C-N axially chiral

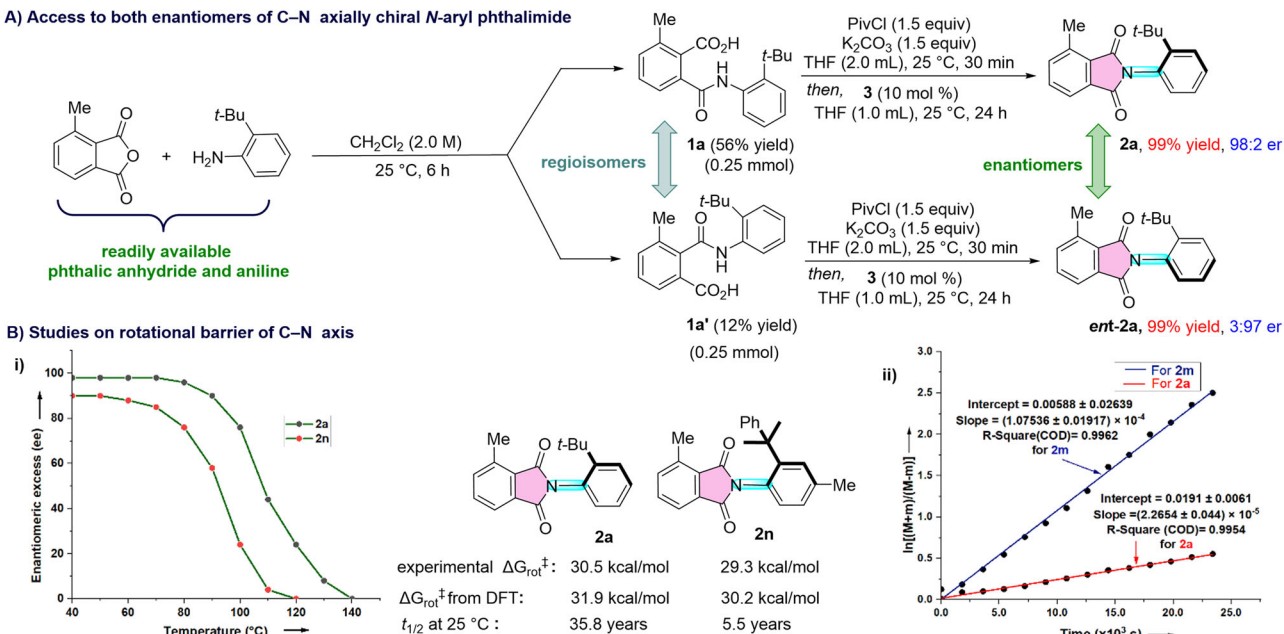

**Fig. 3 | Synthesis of both enantiomers using the same NHC catalyst and studies on C-N rotational barrier. A** Accessing both enantiomers of the product from the same phthalic anhydride and aniline. **B** studies on rotation barrier: (i) effect of C-N bond rotation on temperature, (ii) plot for calculating the rotational barrier (M and m represent the percentage of major and minor enantiomer).

### A) Scale-up synthesis of *N*-aryl phthalimides

### B) Synthesis of *N*-aryl phthalimides in low catalyst loading

**Fig. 4 | Practicality of the developed protocol. A** Scale-up synthesis of *N*-aryl phthalimides. **B** Low catalyst loading synthesis of *N*-aryl phthalimides.

### A) Isolation of isoimide intermediate

### B) Reaction with isoimide intermediate

**Fig. 5 | Mechanistic experiments. A** Isoimide intermediate isolation. **B** Reaction using isoimide intermediate.

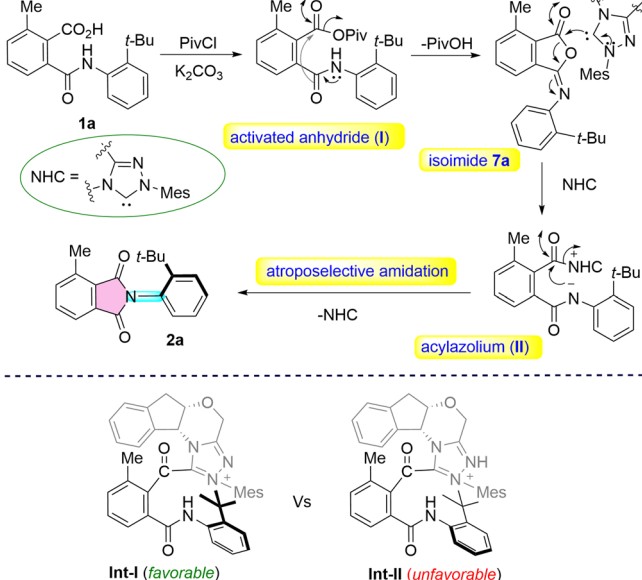

**Fig. 6 | Tentative mechanism of the reaction and the envisioned mode of enantioinduction.**

benzyl bromide derivative **10a** (Fig. 8). Oxidation of **10a** in presence of IBX and DMSO afforded the axially chiral aldehyde **11a** in 93% yield and 97:3 er. An alkene moiety was successfully installed via the Wittig reaction using the aldehyde **11a**, which afforded the styrene derivative **12a** in 66% yield and 94:6 er. The nucleophilic substitution of bromide in **9a** by using NaN₃ furnished the benzyl azide **13a** in 94% yield and 98:2 er. Interestingly, the (3 + 2) cycloaddition of the azide **13a** with the aryne generated from the TMS-triflate precursor under transition-metal-free conditions resulted in the synthesis of the benzotriazole derivative **14a** in 95% yield and 98:2 er. Regarding the functionalization of the *N*-aryl maleimides, monobromination of the double bond in **9a** was accomplished using bromine and the desired product **9f** was

formed in 92% yield and 92:8 er. The bromo derivative **9f** underwent aza-Michael reaction with *n*-butyl amine to form the C-N axially chiral amino maleimide **15a** in 41% yield and 95:5 er. Finally, hydrogenation of **9a** using Pd/C afforded the C-N axially chiral *N*-aryl succinimide **16a** as a single diastereomer bearing both point and axial chirality.

## Discussion

In conclusion, we have demonstrated NHC-catalyzed atroposelective synthesis of *N*-aryl phthalimides and maleimides by the traditional N-$C_{C=O}$ disconnection approach employing the acid activation strategy. By using the conventional and simple disconnection approach for *N*-aryl phthalimides, the reaction proceeds under much milder conditions, excellent yields, and high enantioselectivities, and is suitable for lower catalyst loadings[65]. Interestingly, using the same enantiomer of NHC pre-catalyst, both enantiomers of the product could be accessed starting from same phthalic anhydride and aniline. Preliminary mechanistic investigation unveils that PivCl triggers the activation of phthalamic acid, culminating in the formation of a reactive

isoimide intermediate. Subsequent introduction of NHC opens a unique avenue for the generation of the pivotal acylazolium intermediate, marking a crucial step in the process. In addition, the rotational energy barrier for the C-N bond in *N*-aryl phthalimides were calculated experimentally and using DFT study concluding that the products have configurationally stable C-N chiral axis. The derivatized axially chiral phthalimides/maleimides have proven the synthetic utility of the present study.

## Methods

### Procedure for the atroposelective synthesis of N-Aryl phthalimides

In a flame-dried screw-capped test tube equipped with a magnetic stir bar was taken $K_2CO_3$ (52 mg, 0.375 mmol, 1.5 equiv) from the glovebox, then phthalamic acid derivative **1** (0.25 mmol, 1.0 equiv) was added. Then, the screw-capped tube was evacuated and backfilled with argon. To this mixture was added THF (2.0 mL) under argon atmosphere followed by PivCl (43 µL, 0.375 mmol, 1.5 equiv). The resultant reaction mixture was kept stirring at 25 °C until the full conversion of acid to the corresponding anhydride (monitored by TLC; typically, 30 minutes). To this mixture, the triazolium salt **3** (9.2 mg, 0.025 mmol, 10 mol %) and THF (1.0 mL) were successively added and stirred for 24 h. Then the solvent was evaporated to get the crude residue, which was purified by flash column chromatography on silica gel to afford the corresponding C-N axially chiral phthalimides.

### Procedure for the atroposelective synthesis of N-Aryl maleimides

In a flame-dried screw-capped test tube equipped with a magnetic stir bar was taken $K_2CO_3$ (52 mg, 0.375 mmol, 1.5 equiv) from the glovebox, then carboxylic acid derivative **8** (0.25 mmol, 1.0 equiv) was added. Then, the screw-capped tube was evacuated and backfilled with argon. To this mixture was added THF (1.0 mL) under argon atmosphere followed by PivCl (43 µL, 0.375 mmol, 1.5 equiv). The resultant reaction mixture was kept stirring at 25 °C until the full conversion of acid to the corresponding anhydride (monitored by TLC; typically, 30 minutes). To this mixture, the triazolium salt **3** (9.2 mg, 0.025 mmol, 10 mol %) and THF (0.5 mL) were successively added and stirred for 36 h. Then the solvent was evaporated to get the crude residue, which was purified by flash column chromatography on silica gel to afford the corresponding C-N axially chiral maleimides.

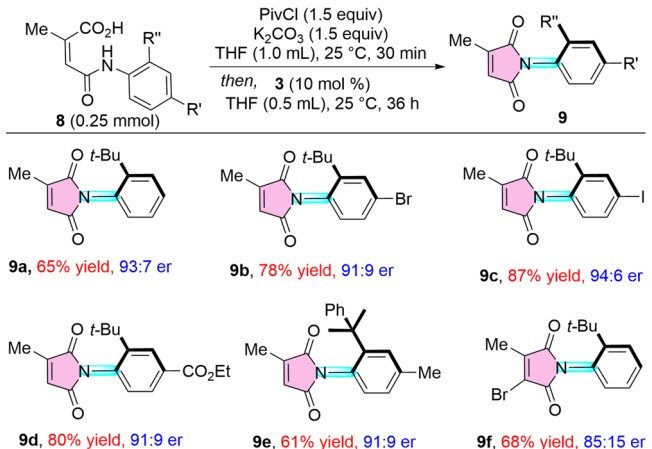

**Fig. 7 | Scope for the synthesis of C-N axially chiral maleimides.** General conditions: **8** (0.25 mmol), **3** (10 mol %), PivCl (1.5 equiv), $K_2CO_3$ (1.5 equiv), THF (1.5 mL), 25 °C, and 36 h. Yields of isolated products are given and the er was established by HPLC analysis on a chiral stationary phase.

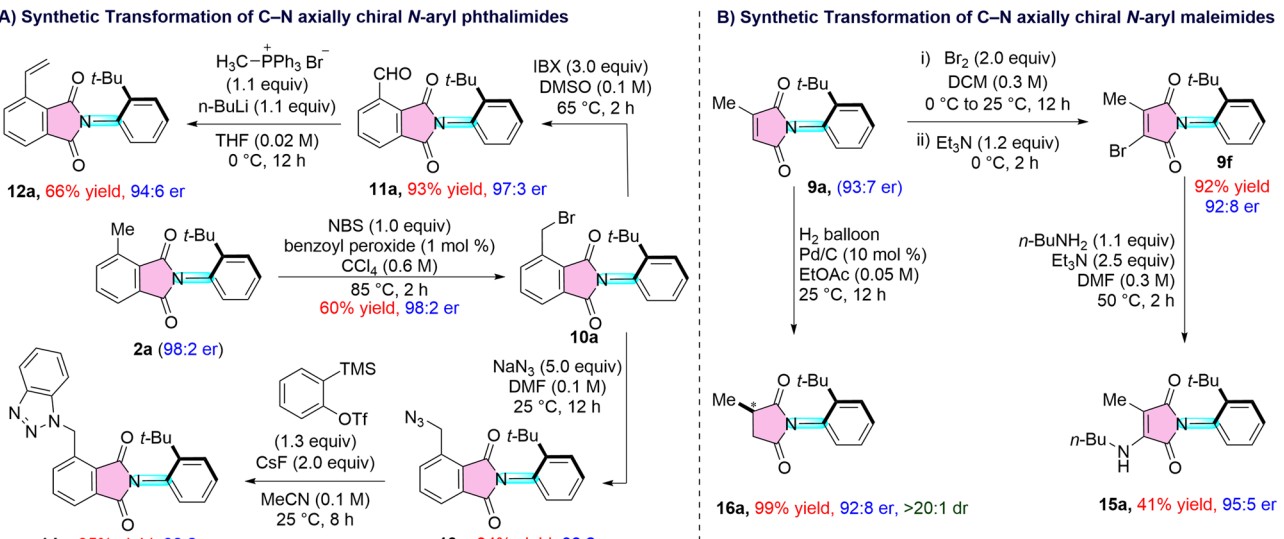

**Fig. 8 | Functionalization of C-N axially chiral phthalimides and maleimides. A** Functionalization of C-N axially chiral *N*-aryl phthalimides. **B** Functionalization of C-N axially chiral *N*-aryl maleimides.

**Procedure for the racemic synthesis of N-Aryl phthalimides/ maleimides**

In a flame-dried screw-capped test tube equipped with a magnetic stir bar was taken $K_2CO_3$ (21.0 mg, 0.15 mmol, 1.5 equiv) from the glove-box, then carboxylic acid derivative **1/9** (0.1 mmol, 1.0 equiv) was added. Then, the screw-capped tube was evacuated and backfilled with argon. To this mixture was added THF (0.8 mL) under argon atmosphere followed by PivCl (18 μL, 0.15 mmol, 1.5 equiv). The resultant reaction mixture was kept stirring at 25 °C until the full conversion of acid to the corresponding anhydride (monitored by TLC; typically, 30 minutes). To this mixture, the triazolium salt **17** (2.7 mg, 0.01 mmol, 10 mol %) was added and stirred for 24 h. Then the solvent was eva-porated to get the crude residue, which was purified by column chromatography on silica gel to afford the corresponding race-mic phthalimides/maleimides.

## Data availability

The authors state that the data supporting the findings of this study are available within the article and Supplementary Information file, or from the corresponding author upon request. The data for the coor-dinates of the optimized structures are present in the Source Data file in the Supplementary Information Section. Crystallographic data for the structures reported in this article have been deposited at the Cambridge Crystallographic Data Center, with the deposition numbers CCDC 2252546 (**2w**), CCDC 2252545 (**2x**), CCDC 2261808 (**1a**) and CCDC 2261807 (**1w**). These data can be obtained free of charge from Data Center via www.ccdc.cam.ac.uk/data_request/cif Source data are provided with this paper.

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

## Acknowledgements

Financial support by the Science and Engineering Research Board (SERB), Government of India, (File Number: CRG/2021/001803) is greatly acknowledged. S.B. thanks MHRD (for PMRF) and S.S.R. thank UGC for fellowships. We thank Prof. Garima Jindal and Mr. Mahesh Singh Harariya for the calculation of the C-N rotational barrier using DFT studies. We thank Dr. Kuruva Balanna, Mr. Sayan Shee, Dr. Avishek Guin and Mr. Sukriyo Chakraborty for helpful discussions.

## Author contributions

S.B. and A.T.B. conceived and designed the project. S.B. performed the optimization studies, substrate scope analysis and mechanistic studies. S.S.R. helped in the substrate scope studies. S.B. wrote the first draft of the manuscript and A.T.B. edited the manuscript. All authors have given approval to the final version of the manuscript.

## Competing interests

The authors declare no competing interests.
