## [Peer Review File · Nature Communications]

N Heterocyclic Carbene-Catalyzed Atroposelective Synthesis of N-Aryl Phthalimides and Maleimides via Activation of Carboxylic AcidsReviewers' comments:

Reviewer #1 (Remarks to the Author):

The manuscript submitted by Biju and colleagues describes a mild methodology for the organocatalyzed atroposelective synthesis of N-aryl phthalimides, which can also be extended to the atroposelective synthesis of N-aryl maleimides. The two-step synthetic approach consists firstly of the synthesis of phthalimidic acid intermediates by the action of an ortho-substituted aniline on a phthalic anhydride, which is then activated by pivaloyl chloride in the presence of an NHC-type catalyst, to lead to the desired product by an intramolecular amidation reaction of an acylazolium-type intermediate. The synthetic approach to the control of axial chirality around a C-N axis is original, even if it still fits in with a certain continuity of the team's work in this theme, as attested by the numerous self-citation publications in this manuscript.

The experimental work is carried out with seriousness and scientific rigor, the article is written with great clarity, and the effectiveness of the method in terms of yield and enantioselectivity control is undeniable. This contribution completes the state-of-the-art in the enantioselective synthesis of C-N atropisomers. Nevertheless, it is aimed at a more specialized audience in organic synthesis than the very general one of Nature Communications, for the following reasons.

- First of all, the study is largely focused on the synthesis of N-aryl phthalimide derivatives, but it is much less illustrated on the synthesis of maleimides, which may be a first limitation. While the approach of controlling axial chirality is original for the atroposelective synthesis of phthalimides, the fact remains that these products can be prepared with comparable efficiency by other already published routes, as attested by the publications cited in the article by the authors.
- Apart from two or three further examples, the scope of the reaction is limited to the use of the tertiary butyl substituent in the ortho position of the aryl group. Is it possible to introduce other substituents in this position? What about enantioselectivity if two substituents of different nature and steric hindrance are present in the ortho and ortho' positions of the aromatic ring?
- Finally, the major problem is that the first step of the synthesis, i.e. the reaction between an ortho-substituted aniline and a phthalic anhydride, leading to the corresponding phthalic acid derivative, is not regioselective, and therefore requires liquid chromatographic separation of the two regioisomers before the second step can be implemented.

In this context and in view of these limitations, this manuscript describes a synthetic method that is efficient in terms of stereoselectivity control but does not really provide a new concept for the synthesis of this type of products. As this criterion of originality is essential for publication in Nature Comm, the present manuscript is not suitable for publication in this journal.

I recommend that this paper be submitted to journals more specialized in organic synthesis as is, with just one minor correction, which consists in adding H₂ to the reaction conditions for the transformation of 9a into 16a, in Scheme 8.

Reviewer #2 (Remarks to the Author):

Biju and coworkers report an atroposelective construction of N-aryl phthalimides and maleimides via N-heterocyclic carbene-catalyzed activation of carboxylic acids under mild conditions. The corresponding products were obtained with good enantioselectivities and the method looks robust and functional group tolerant. In addition the authors also showed that both enantiomers of products can be obtained from the same starting materials using the same NHC catalyst but through different intermediates. Furthermore, the authors disclosed a new mode of the generation of acylazoliums. The nice work has been well conducted with sufficient details, and can be of utility to researchers interested in the organic chemistry and medicinal chemistry. Therefore, this reviewer recommend to accept it after some minor revisions.

1. In references 26-41, the authors mentioned some examples on the NHC-activation mode. Nevertheless, some other recent reviews on this topic are suggested to be included in these references. For example, *Sci. China Chem*, 2022, 65, 1691–1703 ; *Org. Chem. Front.*, 2022, 9, 5016-5040.

2. Did the authors try non-benzoic anydride (e.g. 2-methylsuccinic anydride)?

3. How about the six-membered cyclic anydride (e.g. homophthalic anydride)?

4. Did the authors try other types of alkyl substituted NHC catalysts (2,4,6-triisopropylphenyl instead of Mes in cat.3 should be tested)?

5. General procedure for preparation of the 2/9 and corresponding racemic samples should be included in the text.

6. In the reference 65, the authors mentioned that the poor er values of the products was due to their low C-N rotational barrier, the authors should provide their ΔG_{rot}^\ddagger from DFT

Reviewer #3 (Remarks to the Author):

Biju and co-workers report NHC-catalyzed atroposelective synthesis of N-aryl phthalimides and maleimides by employing the acid activation strategy. Under much milder conditions, a series of target compounds were obtained in high yields with moderate to high enantioselectivities. A proposed mechanism of the reaction was given. Interestingly, using the same enantiomer of NHC pre-catalyst, both enantiomers of the products could be accessed starting from same phthalic anydrides and anilines. However, authors conclude that the products was obtained in excellent enantioselectivities and reaction is proceeding under lower catalyst loading. It was regret that some N-aryl phthalimides and all maleimides were afforded in moderate enantioselectivities in the presence of 10 mol % catalyst. I cannot recommend the manuscript for publication in present form.

Here are some concerns of the referee:

1. In fact, the general procedure for the atroposelective synthesis of N-aryl phthalimides/maleimides was an 'one-pot' reaction involving two-step process. So, all the

Schemes in the manuscript and Supporting Information should be revised.

2. In order to enlarge the scope of the substituted groups in substrates, other electron-withdrawing groups or electron-donating groups at the 3-, 4- or 5-position of the phthalamic acids should be tested.

3. Page 7: 'Moreover, lowering the catalyst loading to 2.0 mol % did not have any adverse implications on the reaction outcome. Using 2.0 mol % of 3, the desired products 2a and 2t were synthesized in comparable yields and selectivities (Scheme 4B), thereby demonstrating the practicality of the developed protocol.' However, the yields and enantioselectivities of the products decreased obviously.

4. Scheme 6: The complete electron transfer process should be presented in the tentative mechanism of the reaction.

5. Scheme 8: The compound 12a was afforded in 66% yield with 88% ee and the ee value decreased obviously. I suggest that better synthetic method should be tested.

6. The dr value of compound 16a should be provided.

Referee 1

We thank the Referee for evaluating our manuscript. The comments of this Referee have helped us a lot in improving the scope of the reaction and hence the quality of the work by performing additional experiments. Our point-by-point response to the comments of this referee is pasted below.

Comment 1: *The manuscript submitted by Biju and colleagues describes a mild methodology for the organocatalyzed atroposelective synthesis of N-aryl phthalimides, which can also be extended to the atroposelective synthesis of N-aryl maleimides. The two-step synthetic approach consists firstly of the synthesis of phthalimidic acid intermediates by the action of an ortho-substituted aniline on a phthalic anhydride, which is then activated by pivaloyl chloride in the presence of an NHC-type catalyst, to lead to the desired product by an intramolecular amidation reaction of an acylazolium-type intermediate. The synthetic approach to the control of axial chirality around a C-N axis is original, even if it still fits in with a certain continuity of the team's work in this theme, as attested by the numerous self-citation publications in this manuscript.*

Response 1: We thank the Referee for the valuable comment on our manuscript. We appreciate the Referee for indicating that the synthetic approach presented in the manuscript is original. Regarding the Referee's comment on numerous self-citations, we have cited only 4 references from our previous work, which include:

Ref 51: NHC-catalyzed activation of carboxylic acids

Ref 57: Desymmetrization strategy for C-N axially chiral succinimides using NHC catalysis.

Ref 60: Kinetic Resolution strategy for the synthesis of C-N axially chiral maleimides.

Ref 62: NHC-catalyzed routes to N-N axially chiral quinazolinones

We believe that all these references are important in the context of the present work.

Comment 2: *The experimental work is carried out with seriousness and scientific rigor, the article is written with great clarity, and the effectiveness of the method in terms of yield and enantioselectivity control is undeniable. This contribution completes the state-of-the-art in the enantioselective synthesis of C-N atropisomers. Nevertheless, it is aimed at a more specialized audience in organic synthesis than the very general one of Nature Communications, for the following reasons.*

Response 2: We thank the Referee for the extremely positive comments on our manuscript. In spite of these positive comments, we do not understand why the Referee is rejecting this paper in the journal. Please note that the submitted work is the first report on NHC-catalyzed method for the synthesis of C-N axially chiral phthalimides/maleimides. Although the conversion of phthalic anhydrides to phthalimides are very well known, the enantioselective version is unknown as there is no chiral center created. In our method, we created the C-N axis via the NHC-catalyzed unique activation of carboxylic acids.

Comment 3: *First of all, the study is largely focused on the synthesis of N-aryl phthalimide derivatives, but it is much less illustrated on the synthesis of maleimides, which may be a first limitation. While the approach of controlling axial chirality is original for the atroposelective synthesis of phthalimides, the fact remains that these products can be prepared with comparable efficiency by other already published routes, as attested by the publications cited in the article by the authors.*

Response 3: We thank the Referee for this comment. In addition to the synthesis of C-N axially chiral phthalimides, we have presented 6 examples of maleimides. With the yield and selectivity values in C-N axially chiral maleimide scope, it is reasonable that the scope of the reaction can easily be extended.

In response to the Referee's query on related C-N axially chiral phthalimide synthesis, please note that there is only one organocatalytic route for the enantioselective synthesis of

phthalimides (*Org. Lett.* **2022**, *24*, 8300-8304; ref 24 of the manuscript). Please note that the benzannulation works only with nitro-substituted maleimides only (pre-functionalized substrates needing harsh conditions for synthesis), and scope of the reaction is very narrow. We believe that our method using NHC-catalysis is broader with 30 examples.

Comment 4: *Apart from two or three further examples, the scope of the reaction is limited to the use of the tertiary butyl substituent in the ortho position of the aryl group. Is it possible to introduce other substituents in this position? What about enantioselectivity if two substituents of different nature and steric hindrance are present in the ortho and ortho' positions of the aromatic ring?*

Response 4: We thank the Referee for this comment. As suggested by this Referee, we have prepared two more examples replacing the *tert*-butyl group with the diaryl alkoxy groups, where the yields and selectivity are good. Hence, we have a total of 5 examples replacing the *tert*-butyl group with other groups.

Based on the suggestion of this Referee, we have prepared di-*ortho* substituted aniline and the corresponding phthalamic acids. The synthesized phthalamic acids under the optimized reaction

solvent	temp (°C)	time (h)	conclusion
CH ₂ Cl ₂ (2.0 mL)	40	12	Both A&B unreacted
CHCl ₃ (2.0 mL)	65	12	Both A&B unreacted
toluene (2.0 mL)	110	24	Both A&B unreacted
AcOH (2.0 mL)	25	12	Both A&B unreacted
AcOH (2.0 mL)	120	36	Both A&B unreacted

conditions afforded the corresponding phthalimides with excellent yields but poor enantioselectivities. Even lowering the temperature to 0 °C, the enantioselectivity did not improve.

Moreover, we have synthesized the 2-(*tert*-butyl)-4,6-dichloroaniline and 2,4-dibromo-6-(*tert*-butyl)aniline. However, all our efforts to synthesize the corresponding phthalamic acid failed. The conditions we tried are summarized in the below Table. It is reasonable to assume that the aniline is surrounded by two bulky groups at the *ortho* positions, making the aniline nitrogen less nucleophilic for attacking to the phthalic anhydride.

The information that the di-*ortho* substituted substrates either failed to provide products with good er values or the failure of the substrate preparation has been indicated in the manuscript (please see the footnote 66 of the manuscript).

Comment 5: Finally, the major problem is that the first step of the synthesis, i.e. the reaction between an *ortho*-substituted aniline and a phthalic anhydride, leading to the corresponding phthalic acid derivative, is not regioselective, and therefore requires liquid chromatographic separation of the two regioisomers before the second step can be implemented.

Response 5: We thank the Referee for this comment. Please note that in the majority of the phthalamic acid synthesis, the product formation is regioselective (the amine addition to the carbonyl, which is less sterically crowded). In the parent substrate, the regioselective phthalamic acids are prepared in 5:1 regioselectivity. Kindly note that in the case of substituted phthalamic acids, the regioselectivity is better and the column purification is easier.

Comment 6: In this context and in view of these limitations, this manuscript describes a synthetic method that is efficient in terms of stereoselectivity control but does not really provide a new concept for the synthesis of this type of products. As this criterion of originality is essential for publication in *Nature Comm*, the present manuscript is not suitable for publication in this journal.

I recommend that this paper be submitted to journals more specialized in organic synthesis as is, with just one minor correction, which consists in adding H₂ to the reaction conditions for the transformation of 9a into 16a, in Scheme 8.

Response 6: We thank the Referee for this comment. Typically, N-aryl phthalimides are prepared from phthalic anhydride and anilines. Although this method is several years old, no enantioselective version is known for obvious reasons as there is no point chiral center created. By using 2-*tert* butyl anilines, we have shown that we could synthesize N-aryl phthalimides with the C-N axis with high yields and ee values. We sincerely hope that the Referee will agree with the novelty of the manuscript. Please note that there is only one organocatalytic enantioselective version known for phthalimide synthesis (*Org. Lett.* 2022, 24, 8300). This report has limitations in terms of substrate scope and works only with 5-nitro 2-*tert*-butyl maleimides and has only 14 examples. We sincerely hope that the Referee will understand the merit of our work over related works.

Referee 2

We express our gratitude to the Referee for the very positive comments and recommending publication of this work **after minor revisions**. The comments of this Referee have helped us a lot in improving the quality of the work by performing additional experiments. Our point-by-point response to the comments of this referee is pasted below.

General Comment:

Biju and coworkers report an atroposelective construction of N-aryl phthalimides and maleimides via N-heterocyclic carbene-catalyzed activation of carboxylic acids under mild conditions. The corresponding products were obtained with good enantioselectivities and the method looks robust and functional group tolerant. In addition the authors also showed that both enantiomers of products can be obtained from the same starting materials using the same NHC catalyst but through different intermediates. Furthermore, the authors disclosed a new mode of the generation of acylazoliums. The nice work has been well conducted with sufficient details, and can be of utility to researchers interested in the organic chemistry and medicinal chemistry. Therefore, this reviewer recommend to accept it after some minor revisions.

Response : We thank this Referee for the insightful and positive comment on our manuscript.

Our responses to the specific comments of this referee include:

Comment 1: *In references 26-41, the authors mentioned some examples on the NHC-activation mode. Nevertheless, some other recent reviews on this topic are suggested to be included in these references. For example, Sci. China Chem, 2022, 65, 1691–1703 ; Org. Chem. Front., 2022, 9, 5016-5040.*

Response 1: We thank the Referee for this suggestion. We have included the suggested recent reviews on this topic in References. Please see References 26-27 (in page 14).

Comment 2: *Did the authors try non-benzoic anhydride (e.g. 2-methylsuccinic anhydride)?*

Response 2: We thank the Referee for this query. Yes, we have tried the reaction with 2-methylsuccinic anhydride, however when 2-methylsuccinic anhydride was treated with 2-*tert* butyl aniline the corresponding acid was not formed in all the conditions we tried.

Comment 3: *How about the six-membered cyclic anhydride (e.g. homophthalic anhydride)?*

Response 3: We thank the Referee for this insightful query. Based on the suggestion of Referee, we tried the reaction of homophthalic anhydride with 2-*tert* butyl aniline in CH₂Cl₂ and the reaction resulted in the formation of the corresponding carboxylic acid derivative in 51% yield.

Unfortunately, when the carboxylic acid was treated under the optimized reaction conditions, the corresponding homophthalic anhydride was not formed.

Comment 4: *Did the authors try other types of alkyl substituted NHC catalysts (2,4,6-triisopropylphenyl instead of Mes in cat.3 should be tested)?*

Response 4: We thank the Referee for this comment. Since the Bode catalyst **3** was giving 99% yield and 98:2 er value, we proceeded the substrate scope with **3** as precatalyst. Based on the comment of this Referee, we have tried the reaction using the 2,4,6-triisopropylphenyl instead of Mes in **3**, but the results are not encouraging.

Comment 5: *General procedure for preparation of the 2/9 and corresponding racemic samples should be included in the text.*

Response 5: We thank the Referee for this suggestion. We have included the General procedure for preparation of the **2/9** and corresponding racemic samples in the manuscript please see page 11-12 in manuscript.

Comment 6: *In the reference 65, the authors mentioned that the poor er values of the products was due to their low C-N rotational barrier, the authors should provide their ΔG_{rot}^\ddagger from DFT*

Response 5: We thank the Referee for this insightful suggestion. Based on the suggestion of this Referee, we have calculated the rotation barrier of **2af** from DFT. The ΔG_{rot}^\ddagger from DFT is 35.5 kcal/mol, which is higher than the parent compound **2a**. So, the origin of low enantioselectivity may not be because of low C-N rotational barrier as we have envisioned earlier. It is reasonable to believe that that during the atroposelective amidation, the catalyst can't distinguish the rather small steric difference between two ortho-substituents. So, both the

enantiomers were formed in equal ratio. Because of this new result, we have modified the reference 65, which read as "The reactions performed using di-ortho substituted aniline-derived phthalamic acid substrates although reacted well but with poor er values."

Referee 3

We thank this Referee for the positive and insightful comments on our manuscript. The comments of this Referee have helped us a lot in improving the quality of the work by performing additional experiments. Our point-by-point response to the comments of this referee is pasted below.

General Comment : *Biju and co-workers report NHC-catalyzed atroposelective synthesis of N-aryl phthalimides and maleimides by employing the acid activation strategy. Under much milder conditions, a series of target compounds were obtained in high yields with moderate to high enantioselectivities. A proposed mechanism of the reaction was given. Interestingly, using the same enantiomer of NHC pre-catalyst, both enantiomers of the products could be accessed starting from same phthalic anhydrides and anilines. However, authors conclude that the products was obtained in excellent enantioselectivities and reaction is proceeding under lower catalyst loading. It was regret that some N-aryl phthalimides and all maleimides were afforded in moderate enantioselectivities in the presence of 10 mol % catalyst. I cannot recommend the manuscript for publication in present form.*

Response : We thank this referee for the insightful comment on our manuscript. Please note that in organocatalytic transformations, use of 10-20 mol % catalyst for a given transformation is normal. In our case, lowering the NHC precatalyst loading to 2.0 mol % did not affect the reactivity much (please see Scheme 4B). Sorry for the confusion if any happened.

Our responses to the specific comments of this Referee include:

Comment 1: *In fact, the general procedure for the atroposelective synthesis of N-aryl phthalimides/maleimides was an 'one-pot' reaction involving two-step process. So, all the Schemes in the manuscript and Supporting Information should be revised.*

Response 1: We express our gratitude to the Referee for this valuable and insightful suggestion. Based on the suggestion of this Referee, we have modified all the Schemes in the manuscript and Supporting Information.

Comment 2: *In order to enlarge the scope of the substituted groups in substrates, other electron-withdrawing groups or electron-donating groups at the 3-, 4- or 5-position of the phthalamic acids should be tested.*

Response 2: We thank the Referee for this suggestion. We have tried to introduce several electron-withdrawing and -donating group in the 3,4- position of benzoic acid moiety, but we were unsuccessful to achieve the corresponding phthalamic acid substrate for the catalytic reaction.

However, to expand the substrate scope as suggested by this Referee, we have synthesized three more examples. The new entries are:-

These examples not only expand the substrate scope but also indicate that the incorporation of di-aryl or di-heteroaryl moieties effectively restrains rotation around the C-N bond, thereby facilitating the formation of **2p** and **2q** with high enantioselectivity.

Comment 3: Page 7: 'Moreover, lowering the catalyst loading to 2.0 mol % did not have any adverse implications on the reaction outcome. Using 2.0 mol % of **3**, the desired products **2a** and **2t** were synthesized in comparable yields and selectivities (Scheme 4B), thereby demonstrating the practicality of the developed protocol.' However, the yields and enantioselectivities of the products decreased obviously.

Response 3: We thank the Referee for this suggestion. The corresponding sentence has been modified.

Comment 4.: Scheme 6: The complete electron transfer process should be presented in the tentative mechanism of the reaction.

Response 4: We thank the Referee for this valuable suggestion. We have modified the Schemes where mechanisms have been shown. Please see Scheme 1 in page 2 and Scheme 6 in page 9.

Comment 5: Scheme 8: The compound **12a** was afforded in 66% yield with 88% ee and the ee value decreased obviously. I suggest that better synthetic method should be tested.

Response 5: We thank the Referee for this suggestion. We also tried the other conditions for the Wittig reaction by varying the bases, the results are not better compared to the use of *n*-BuLi as the base.

Comment 6: The *dr* value of compound **16a** should be provided.

Response 6: We appreciate the Referee for this suggestion. The *dr* value for **16a** is >20:1, which is now mentioned in the Scheme. Please see Scheme 8 in page 11.

In view of the modifications in the manuscript as indicated, I hope you will consider this manuscript for publication in *Nature Communications*.

I would be glad to provide any other additional information.

REVIEWER COMMENTS

Reviewer #2 (Remarks to the Author):

The authors have almost fully revised this manuscript according to all referee's suggestions. This reviewer believes that the current version meets the standard of Nature Communications.

Reviewer #3 (Remarks to the Author):

The manuscript has been mostly revised according to the review comments. However, the supporting information has been partially revised. There are still many errors and it should be checked carefully!

For example:

The title in supporting information is inconsistent with the manuscript.

Page S6, Procedure for the Low Catalyst Loading: The amount of catalyst 3 (10 mol%) in the Scheme is wrong.

Page 78, Product Functionalization: The scheme B) Synthetic Transformation of C–N axially chiral N-aryl maleimides' and the corresponding experimental procedure.

Our point-by-point responses to the comments of the Referees are provided below:

Reviewer 2

Comment: *The authors have almost fully revised this manuscript according to all referee's suggestions. This reviewer believes that the current version meets the standard of Nature Communications.*

Response : We express our gratitude to the Referee for the very positive comments and recommending publication of this work in *Nature Communications* in the present form.

Reviewer 3

We express our gratitude to the Referee for the very positive comments and recommending publication of this work after minor modifications in the Supporting Information.

General Comment: *The manuscript has been mostly revised according to the review comments. However, the supporting information has been partially revised. There are still many errors, and it should be checked carefully!*

Response : We thank this Referee for the very positive comment on our manuscript and Supporting Information. The Supporting Information has been checked carefully and the modification has been done according to the suggestion of Referee.

Our responses to the specific comments of this Referee include:

Comment 1: *The title in supporting information is inconsistent with the manuscript.*

Response 1: We thank the Referee for this comment. We are sorry for the mistake. The title of the Supporting Information has been changed now. We also did minor modification in the title of the manuscript, and the same is now reflecting in the Supporting Information.

Comment 2: *Page S6, Procedure for the Low Catalyst Loading: The amount of catalyst 3 (10 mol%) in the Scheme is wrong.*

Response 2: We thank the Referee for this comment. We are sorry for this mistake. We have changed the amount of the catalyst **3** to 2.0 mol %.

Comment 3: *Page 78, Product Functionalization: The scheme B) Synthetic Transformation of C–N axially chiral N-aryl maleimides' and the corresponding experimental procedure.*

Response 3: We thank the Referee for this comment. We are sorry for this mistake. We have changed the Scheme B in page S78 and the corresponding experimental procedure.